# Molecular Dysregulation in Autism Spectrum Disorder

**DOI:** 10.3390/jpm11090848

**Published:** 2021-08-27

**Authors:** Pritmohinder S. Gill, Jeffery L. Clothier, Aravindhan Veerapandiyan, Harsh Dweep, Patricia A. Porter-Gill, G. Bradley Schaefer

**Affiliations:** 1Department of Pediatrics, University of Arkansas for Medical Sciences, Little Rock, AR 72202, USA; SchaeferGB@uams.edu; 2Arkansas Children’s Research Institute, 13 Children’s Way, Little Rock, AR 72202, USA; PortergillPA@archildrens.org; 3Psychiatric Research Institute, University of Arkansas for Medical Sciences, Little Rock, AR 72205, USA; JLClothier@uams.edu; 4Pediatric Neurology, Arkansas Children’s Hospital, 1 Children’s Way, Little Rock, AR 72202, USA; AVeerapandiyan@uams.edu; 5The Wistar Institute, 3601 Spruce St., Philadelphia, PA 19104, USA; hdweep@Wistar.org; 6Genetics and Pediatrics, University of Arkansas for Medical Sciences, Little Rock, AR 72202, USA; 7Arkansas Children’s Hospital NW, Springdale, AR 72762, USA

**Keywords:** autism spectrum disorder (ASD), genetic, copy number variation (CNV), epigenetic, knockout models, endophenotypes, pharmacogenomics, biomarker

## Abstract

Autism Spectrum Disorder (ASD) comprises a heterogeneous group of neurodevelopmental disorders with a strong heritable genetic component. At present, ASD is diagnosed solely by behavioral criteria. Advances in genomic analysis have contributed to numerous candidate genes for the risk of ASD, where rare mutations and s common variants contribute to its susceptibility. Moreover, studies show rare de novo variants, copy number variation and single nucleotide polymorphisms (SNPs) also impact neurodevelopment signaling. Exploration of rare and common variants involved in common dysregulated pathways can provide new diagnostic and therapeutic strategies for ASD. Contributions of current innovative molecular strategies to understand etiology of ASD will be explored which are focused on whole exome sequencing (WES), whole genome sequencing (WGS), microRNA, long non-coding RNAs and CRISPR/Cas9 models. Some promising areas of pharmacogenomic and endophenotype directed therapies as novel personalized treatment and prevention will be discussed.

## 1. Introduction

Autism spectrum disorder (ASD) is a group of complex neurodevelopment disorders involving behavioral difficulties and developmental delays that effect social communication and interactions [1]. Over 78 years back, Leo Kanner [2] described clinically the syndrome of early infantile autism in children aged two to eight years. Todate, the genetic heterogeneity of this syndrome has eluded researchers and clinicians to pinpoint the underlying genetic causes of impaired social interactions, restricted communications skills, and unusual repetitive behaviors. According to the Center for Disease Control (www.cdc.gov/ncbddd/autism/data.html, accessed on 2 July 2021), currently, the prevalence of ASD is 1 in 54 children and is more common among boys. Autism is a heterogeneous disorder with a high heritability [3,4]. A meta-analysis of twin studies on ASD show high heritability for monozygotic twins (MZ), with estimates ranging between 64–91% [5]. This study also suggests that the ASD is due to strong genetic component [5]. Autism etiology has many key players, including common genetic variants, inherited gene-disruptive mutations and rare variants of large effect outside of the coding region [6,7,8]. A number of pathways dysregulated in ASD, involve chromatin remodeling, RNA transcription and splicing, synaptic function, ion-channels, MAPK and calicium signaling [9,10,11,12]. This review will provide current information on innovative technologies like WES, WGS, non-coding RNAs, ASD models, pharmacogenomics and endophenotypes in ASD.

### 1.1. Diagnosis and Epidemiology

ASD is diagnosed by clinicians based on revised criteria of the Diagnostic and Statistical Manual of Mental Disorders V (DSM-V, 2013, ADA) [1]. ASD symptoms are noticeable from early childhood and are associated with neural dysfunctions leading to ASD development. ASD is more common in boys [13] and epidemiologic investigations have established advanced parental age and preterm birth as ASD risk factors [14]. The de novo structural mutations occur at a higher rate in ASD affected individuals as parental age increases [15]. At present, ASD is the most common serious developmental disorder in the USA and the world [13]. The economic impact of ASD is staggering and with increasing prevalence, the costs could reach USD 15 trillion by 2029 [16]. The last decade has seen considerable progress in epidemiological research and, in the near future, a better understanding of ASD etiology is certain with the development of new novel methods.

### 1.2. Multifactorial Inheritance

The occurrence risk pattern of ASD fits multi-factorial traits [17], as it is highly heritable, with contributions from structural variants, common variants, rare inherited alleles and de novo variants [6,9,18,19]. Moreover, other contributors to ASD are chromosomal abnormalities, insertions, deletions, substitutions, and single nucleotide variation (SNV), along with germline, somatic, de novo mutations [7,20,21]. CNVs, such as common and rare, basically points to regions in genome with high and low frequencies of CNV respectively. A large homogeneous Swedish epidemiological sample [6] showed that 14% of affected subjects carry de novo copy number variants (dnCNV) and loss-of-function (LoF) mutations, which account for 2.6% autism liability. These observations suggest that in Sweden, about 60% of genetic variation account for the risk for autism [6]. Another large Swedish cohort study showed that for an individual, the risk of autism is increased 10 fold, if a full sibling has the diagnosis [22].

ASD is also strongly linked with other monogenic mendellian genetic diseases [for example, Fragile X Syndrome (*FMR1*), Tuberous Sclerosis (*TSC1, TSC2*), and Rett Syndrome (*MECP2*)] [23] and account for approximately 3–10% of cases [24]. Mendelian diseases can arise due to genomic rearrangements (e.g., CNV) and produce complex traits such as behaviors, or represent benign polymorphic changes [21] through diverse mechanisms such as gene dosage, gene interruption, generation of a fusion gene, position effects, etc.

## 2. Genetics Studies

Twin and family studies point to genetic basis of ASD susceptibility and is a highly heterogeneous genetic disorder. There are over 100 ASD candidate genes, and Appendix A shows some 58 genes which are involved in transcription, DNA binding, cell growth, post-synaptic density, NMDA glutamate receptor clustering and neuroprotection (Appendix A). Currently, there are no biomarkers at the cellular and molecular level for diagnostic and therapeutic interventions of ASD. ASD is characterized by rare de novo and inherited CNVs which target protein coding genes involved in neuronal development [25]. From a recent Swedish study, it was ascertained that genetic variation accounts for roughly 60% of the variation in risk for autism [6], and rare variants explain a smaller fraction of total heritability compared to common variants [6]. To date, most early published studies, have focused on single-nucleotide polymorphism (SNP) using microarray analysis. Microarrays have their weaknesses which include probe design, high cost and low accuracy, and they only evaluate the identified SNPs in the genome.

### 2.1. Candidate Genes and Linkage Studies

ASD candidate genes play a pivotal role in brain development, as they are involved with key brain structures, neurotransmitters, or neuromodulators [26]. But many of the candidate gene and linkage investigations are fraught with small sample size contributing to low statistical power and fail to replicate findings. A number of studies have reviewed the linkage and candidate gene studies [12,27,28]. Candidate genes which have been the focus of ASD are: *CACNAIC*, *GABAA receptor subunit*, *FOXP2*, *HOXA1*, *HOXB1*, *HTR2A*, *MTHFR*, *RELN*, *RAY1/ST7*, *IMMP2L*, *SLC6A4*, *OXTR*, *UBE3A* and *WNT-2* [26,27,28,29,30,31]. More recently, a role for de novo deleterious *NCKAP1* variants was reported in neurodevelopmental delay/autism, as variants can affect the neuronal migration in early cortical development [32]. *POU3F2*, was identified as ASD risk gene, and it is a key transcription factor involved in neuronal differentiation, whose downstream target genes are strongly enriched for known ASD genes and mutations [33]. 

A autism study identified significant linkage on chromosomes 6q27 (LOD = 2.94) and 20p13 (LOD = 3.81) [34]. To replicate these results, genotyping showed significant association with autism for a SNP on chromosome 5p15 (between *SEMA5A* and *TAS2R1*) and *SEMA5A* showed decreased expression in autism brain tissue [34]. Linkage analysis using 335 markers on 152 families support the evidence for a quantitative trait loci (QTL) for language on chromosome 7q35 [35]. Further, a genome wide linkage screen on 158 multiplex autism families gave evidence of linkage to chromosomes 17p11.2 and 19p13 [36]. 

Candidate and linkage studies show the low throughput nature of these analyses, but also point to the fact that small sample sizes are not enough to understand the complex nature of ASD genetics.

### 2.2. Chromosomal Loci and CNV

To date, more than 50 deletion and duplication syndromes associated with an autism phenotype have been described and sex chromosomal aneuploidies have the highest association with autism [37]. Genetic abnormalities of the chromosome 15q11-q13 region are an important cause for ASD, which account for approximately 1% of cases [38]. ASD susceptibility genes have been localized to various chromosomes, especially 2q, 5p, 7q, 15q, 17q and on chromosome X [16,28,39,40,41]. Structural variation of chromosomes which comprises CNVs including deletion and duplication, translocation, and inversion have a substantial role in ASD [42,43]. Moreover, ASD show association with CNVs that implicate the postsynaptic genes including *SHANK3*, *NLGN4*, and *NRXN1* and protocadherin family member *PCDH9* important in signaling at neuronal synaptic junctions. This study identified two ASD risk loci 15q24 and 16p11.2, which overlap with mental retardation sites [42]. A study on European ancestry samples found numerous de novo and inherited events, sometimes in combination within a given family. This implicates many novel ASD genes including *SHANK2*, *SYNGAP1*, *DLGAP2* and the X-linked DDX53-PTCHD1 locus [44] and the enrichment analysis of CNVs showed involvement in cellular proliferation, and GTPase/Ras signaling. 

Analysis of dnCNVs from the Simons Simplex Collection (SSC) show 71 ASD risk loci, including 6 CNV regions (1q21.1, 3q29, 7q11.23, 16p11.2, 15q11.2-13, and 22q11.2) and 65 risk genes [(FDR ≤ 0.1) [including *NRXN1* and *SHANK3*]] [43]. Recently, a meta-analysis of CNVs from the ‘CNV’ module of Simons Foundation Autism Research Initiative (SFARI) database, identified 105 “prominent CNV regions” specific to ASD, encompassing 537 genes across 56 loci in 20 chromosomes [45]. Rare CNV regions in loci 4p16.3 and 9q34.3 showed the highest cumulative scores and genes within these two CNV loci exhibited the highest neuro-functional networks [45]. A similar analysis from AutDB database reports on eleven genomic loci, along eight chromosomes and covering 166 genes [46]. This study finds the highest CNV burden in ASD subjects on human chromosome 16p11.2 with 27 duplications and 36 deletions. A recent study looked at 6 candidate genes related to ASD, viz. *MTHFR*
*C677T*, *SLC25A12*, *OXTR*, *RELN*, *5-HTTLPR*, *SHANK* [47] and found *MTHFR C667T* variant to be a risk factor for the occurrence of ASD. A meta-analysis for *MTHFR C677T* confirmed it as a susceptibility factor for ASD [48].

In conclusion, CNVs represents alterations of the normal number of gene copies and include deletions, insertions, duplications and complex multi-site variants. The above reviewed studies [42,43,45,46,47,48] show CNVs play an important role in ASD patients and have three to five times more dnCNVs than other family members [42,44].

### 2.3. Genome-Wide Association Studies (GWASs)

Genome-wide association studies (GWAS) have identified potential contributions of common variants of small effect to pathogenesis of ASD [34,49]. GWAS study on an European ancestry cohort of 780 families showed chromosome locus 5p14.1 reached genome wide significance for ASD [50]. Six sSNPs on chromosome 5p14.1 gave strong signals between genes cadherin 10 (*CDH10*) and cadherin 9 (*CDH9*), and with the most significant SNP being rs4307059 [50]. A larger GWAS study on European case-control samples [51] looked at previously reported candidate genes *SEMA5A* (rs10513025) [39], *MACROD2* (rs4141463) [40,52] and *MSNP1* (rs4307059) [50,53] and were unable to replicate these GWAS signals.

A genome-wide significant (GWS) risk loci was detected at 10q24.32 and was associated with social skills [54] and this chromosomal location overlaps with several genes such as *PITX3* which encodes a transcription factor for neuronal differentiation and *CUEDC2* involved in the ubiquitination-proteasomal degradation pathway [54]. This study also found association of ASD with several neurodevelopmental-related genes including *EXT1*, *ASTN2*, ANO4, *MACROD2*, and *HDAC4.* More recently, a unique Danish population resource under iPSYCH project [55], detected five genome-wide significant loci. The study identified 5 index SNPs: rs910805 (Chr 20); rs10099100 (Chr 8); rs201910565 (Chr 1); rs71190156 (Chr 20) and rs111931861 (Chr 7). Notably, a number of genes located in the identified loci have previously been linked to ASD risk in studies of de novo and rare variants including *PTBP2, CADPS*, and *KMT2E*. Moreover, the identified ASD-candidates in this study showed the highest expression during fetal corticogenesis [55]. Gene-based association analysis on primary ASD meta-analysis using MAGMA identified the top associated genes *KIZ* and *XRN2* (Chr 20), *MFHAS1*, *XKR6*, *MSRA*, and *SOX7* (Chr 8). The other associated genes were *KCNN2*, *KANSL1*, *MACROD2*, *WNT3*, *MAPT*, *CRHR1*, *NTM*, *MMP12*, and *BLK* [55]. Summary statistics from this meta-analysis [55] employed PASCAL scoring algorithm with linkage disequilibrium (LD) information retrieved from 1000 Genomes European panel [56], identified the following loci associated with ASD: *XRN2*, *NKX2-4*, *PLK1S1*, *KCNN2*, *NKX2-2*, *CRHR1-IT1*, *C8orf74* and *LOC644172.* Both the analysis by *MAGMA and PASCAL* showed common loci to be *XRN2*, KIZ, and *KCNN2* [55,56]. PASCAL maybe used as a complementary gene-based analysis (GBA) approach as it discovered additional ASD-associated genes.

GWAS approaches [34,40,54,55,56,57], have identified susceptible genes, but innate genetic heterogeneity of ASD has provided a limited success in this application to pinpoint the important variants for diagnostic and therapeutic guidance.

### 2.4. Whole Exome and Genome Sequencing

Recent advances in the development of next-generation sequencing (NGS) technologies provide researchers with unprecedented possibilities for genetic analyses to unravel genetic causes of autism. Pathways of transcriptional regulation and chromatin remodeling are affected by causative mutations in neurodevelopmental disorders (e.g., intellectual disability and autism) [58], and the following landmark approaches are helping to understand the complex nature of ASD.

#### 2.4.1. Whole Exome Sequencing

Whole Exome Sequencing (WES) is an approach to selectively sequence the coding regions (exons) of a genome to uncover rare or common variants associated with a disorder or phenotype [59], as more than 98% of the human genome does not encode protein sequences [60]. WES can help to detect mutations and de novo variants in ASD affected and unaffected individuals. Review of WES studies in ASD, showed involvement of mutations (frameshift, deletion, indel, missense, synonymous, nonsense, splice ste, 3′UTR) in over 100 genes in ASD [61]. By studying a cohort of consanguineous and/or multiplex families with ASD, Yu and co-workers [62] found familial ASD associated with biallelic mutations in disease genes (*AMT, PEX7, SYNE1, VPS13B, PAH, POMGNT1*). Genes implicated here also include ones known to regulate or be regulated by synaptic activity (e.g., *MECP2, SYNE1*). A population-based approach on 933 ASD cases and 869 controls [63], showed that rare autosomal and X chromosome complete gene knockouts are important inherited risk factors for ASD. There was a 2-fold increase in complete knockouts of autosomal genes with low rates of loss of function (LoF) variation (≤5% frequency) in cases and this study observed a significant 1.5-fold increase in rare hemizygous knockouts in males [63].

WES analysis of the Simons Simplex Collection (with 2508 affected children, 1911 unaffected siblings and parents of each family) identified new de novo likely gene-disrupting (LGD) mutations involved in this heterogeneous disorder [19]. This study identified 353 candidate LGD gene targets, and 27 genes recurrently hit by LGD events, which gave credibility to the other studies showing a very complex genetic architecture of ASD. The largest whole exome sequencing study (n = 35,584 total samples, 11,986 with ASD) identified 102 risk genes [64], and most were expressed and enriched early in excitatory and inhibitory neuronal lineages. Of the 102 ASD genes, 60 were not discovered by previous exome sequencing efforts and 30 are considered as “truly novel”, as they have not been implicated in autosomal dominant neurodevelopmental disorders. This study observed a 2-fold enrichment of de novo protein-truncating variants (PTVs) in highly constrained genes in affected females vs. affected males [64]. The result of PTVs in females give credence to female protective model for ASD development. and implies risk variation has larger effects in males than in females. The other salient finding from the results was that among the five GWAS-significant ASD hits [55], KMT2E is implicated by both GWAS and the list of 102 FDR ≤ 0.1 genes [64].

To identify genes with private gene disrupting and missense variants of interest (VOI), Patowary and colleagues [65] looked at 26 families with affected first cousins from the NIMH repository (https://www.nimhgenetics.org/, accessed on 1 August 2021). The genes carrying VOIs were enriched for biological processes related to cell projection organization and neuron development. Missense variants in one gene, *CEP41*, associated significantly with ASD [65] and overexpression of *CEP41* pathogenic alleles in zebrafish model, showed that variants in embryos induces axonal defects and also affects cranial neural crest (CNC) cell migration and exhibited deficits in social behavior. More recently, Kim and co-workers [66] performed WES on 51 Korean families (n = 151 individuals) to identify putative causal variants of ASD and identified 36 de novo variants, which were confirmed by Sanger sequencing (27 missense, two silent, one nonsense, one splice region, one splice site, one 5′ UTR, one intronic SNVs, and two frameshift deletions). A retrospective study on 343 ASD patients using different genetic approaches of *FMR1* testing, chromosomal microarray (CMA) and/or WES, recommend WES as the first-tier approach in the diagnosis of ASD patients [67].

WES studies have extended our in-depth knowledge on the rare de novo SNVs and CNVs impacting protein coding genes leading to dysregulation of signaling pathways controlling nervous system development, neuronal activity, synaptic homeostasis, immune response, chromatin modification, transcription and translation [9,10,11,12].

#### 2.4.2. Whole Genome Sequencing (WGS)

Whole Genome Sequencing (WGS), on the other hand, examines the *whole genome* for SNVs, indels, SV and CNVs in coding and non-coding regions. Non-coding regions cover almost 98% of the human genome [60] and this approach can provide detailed information on cellular and molecular pathways dysregulated in ASD.

To understand the genetic etiology of ASD, Turner and colleagues [68], analyzed the pattern of de novo mutations (DNMs) in 516 autism families (2064 individuals) to investigate the combined effect of genetic and noncoding mutations underlying autism. In their analysis, probands carry more gene-disruptive CNVs and SNVs resulting in severe missense mutations and mapping to predicted fetal brain promoters and embryonic stem cell enhancers. This study observed a twofold enrichment of missense variants and included autism risk genes *PTPN11, CACNA1G, TRIP12*, *PTK7, SUPT16H* and *SCN3A* indicating the importance of particularly severe de novo missense mutations. Moreover, the well-established increase in de novo substitutions with paternal age was observed with strong correlation between the number of de novo SNVs and indels and increase in paternal age; and also for noncoding de novo SNVs and indels [68].

An analysis of whole-genome sequences on 5205 individuals, identified 18 new candidate genes for autism [69], including MED13 and PHF3. Many of the ASD-risk genes identified were enriched in synaptic transmission, transcriptional regulation and RNA processing functions. Moreover, in 11.2% of ASD cases, a molecular basis could be determined and 7.2% of these carried CNV/chromosomal abnormalities [69] highlighting the importance of detecting all forms of genetic variation in ASD for diagnostic and therapeutic interventions. Werling and co-workers [70] observed a median of 64 de novo single nucleotide variants (SNVs) and 5 de novo indels per child across autosomes.No significant enrichments were observed for either de novo or rare inherited structural variants (SVs), though this study detected 171 de novo SVs [70]. Five predicted high-impact variants as de novo were detected in WGS data on 119 individuals [71] along with two novel de novo variants in the ASD gene *SCN2A.*

The emerging technologies such as WES and WGS can provide in-depth information on individuals genome and lead to detection of new ASD genes, including potential diagnostic and therapeutic targets for personalized therapies. Appendix A shows some of the candidate risk genes from all the above-mentioned approaches. The Database for Annotation, Visualization and Integrated Discovery (DAVID v6.8; https://david.ncifcrf.gov/, accessed on 2 August 2021) [72] was used to identify enriched biological themes, particularly Gene Ontology (GO) terms associated with 58 genes (Appendix A). Several genes (*KMT2E, PHF3, KDM5B, KMT2C, CHD8, ILF2, FOXP2, FOXP1, MED13L, MECP2, HOXA1, ADNP, HOXB1, ARID2*) were annotated as transcription regulators under biological processes (BP) category with 2.4-fold more than expected with a *p*-value = 0.004. Similarly, DNA binding, a molecular functions (MF) category, was 2.2-fold more than expected with 11 genes (*MECP2, KDM5B, DEAF1, CHD8, KMT2C, POGZ, ADNP, HOXB1, ARID2, ILF2, FOXP*) and a *p*-value = 0.02. Also, a total of 21 genes (*KMT2E, OXTR, DSCAM, NEGR1, NRXN1, PTEN, ANK2, CACNA1C, HTR2A, SLC6A1, GRIN2B, SLC6A4, RELN, KCNMA1, KCNQ3, CEP41, ARID2, SHANK3, WNT2, SHANK2, SCN1A*) were found to be associated with nucleus cellular component (CC) with 1.9-fold (*p* = 0.002) (Appendix A).

## 3. Epigenetic Studies

Non-coding RNAs (ncRNAs) regulate gene expression at the transcriptional and post-transcriptional level and are important to control epigenetic pathways essential for targeting of histone modifying complexes, chromatin remodeling and DNA methylation. This section will focus more on information on ncRNA especially microRNAs (miRs or miRNAs) and long noncoding RNA (LncRNA), These non-coding RNAs exhibit tissue-specific, cell-specific expression and are important in the development and functioning of the brain.

### 3.1. MicroRNA Studies

miRNAs or miRs are approximately 18–25 nucleotides, noncoding transcripts that control messenger RNA (mRNA) and protein levels by interacting with the 3′ untranslated region (UTR) of specific mRNAs [73]. miRs can control the expression of approximately two-thirds of human mRNAs [74] and 70% of miRs are expressed in the central nervous system (CNS), including the brain and spinal cord [75,76]. As of 2018, there are 38,589 miRNAs that have been discovered (http://www.mirbase.org/, accessed on 17 July 2021 [77]. In addition, several miR studies show that miRs exist in the extracellular fluids like saliva, urine, serum, plasma [78]. These miRs could provide important potential biomarkers for prognostic and diagnostic value; especially serum or plasma due to ease of specimen acquisition.

Over thirty publications were reviewed on miRs and ASD that were published in the past 10–15 years; predominately in the previous 5 years. The focus was on human, not animal or computational, miR manuscripts. Based on the author’s choice, only 15 papers (Appendix A) [78,79,80,81,82,83,84,85,86,87,88,89,90,91,92] were used in this review. For example, in some cases, only a small number of the ASD specimens showed miR differentiation in only a portion of the human study samples. It was also difficult to ascertain whether the study specimens were from adult and/or children in a small number of the studies.

This collection of papers examined miRs differentially expressed in post-mortem brain tissue, lymphocyte and lymphoblastoid cells lines, saliva, serum, and whole blood. Of these 15 manuscripts, specimens were derived from both children and adults.

As for miR technologies used in these miR biomarker discovery papers, whole genome sequencing (WGS), miR arrays, and quantitative RT-PCR were used, and most were validated by a secondary means to suggest a final, smaller number of differentiated miRs expression profiles. Appendix A represents a summary of these finding (study, specimen, number of participants, final result for miRs, and reference). Appendix A shows a heatmap of the miRNAs in ASD (up-regulated vs. down-regulated) from brain, blood lymphocytes, WBC, mature WBC, serum and saliva.

In some cases, the same miRs were found to be expressed in different tissue sources, as well as similar expression patterns were observed (up or down regulation). A few examples of this are the miR-146a, miR-664-3p, miR-151a-3p, and miR-27a-3p (Appendix A and Appendix A). In two separate serum studies, two miRs (486-3p and 328-3p), were down regulated in ASD. In addition, most of the children miR studies were done in easily accessible specimens; saliva, whole blood, and serum. A comparison was made between ASD and miRs in children studies only. Several ASD differentiated miRs overlapped in the children studies and all ASD miR studies in children were age 3–16 years. A study detailing spatio-temporal miRNA expression in the developing human brain [93] showed 75 miRs differentially expressed across this developmental period within different brain regions; prefrontal cortex, hippocampus, and cerebellum. The largest variation occurred between infancy and early childhood. This work [93] also assessed for enrichment of miRNA targets among genes previously implicated in various neurological and psychiatric disorders that have significant genetic etiology, and found that the sex-biased targets were enriched for Wnt signaling and transforming growth factor-beta (TGF-β) [93].

As a single miRNA can target approximately hundreds to thousands of mRNA transcripts and a single mRNA transcript could be targeted by multiple miRNAs [94], so studies dealing with functional miRNA-mRNA interactions in ASD will be of paramount importance.

### 3.2. Long Noncoding RNA (lncRNA) Studies

Long noncoding RNAs (lncRNAs) are the type of non-protein coding transcripts, which are more than 200 nucleotides in length [95,96]. LncRNAs are found throughout the genome and have important roles in cellular functions, by interacting with DNA, RNA and proteins, modulating chromatin structure and function, transcription of neighboring and distant genes, and RNA splicing, stability and translation [97]. Moreover, lncRNAs can regulate protein-coding mRNAs through the mechanism of miRNA sponges [98]. A total of 96,411 genes were generated from 173,112 human transcripts (http://www.noncode.org/, accessed on 2 August 2021) [99]. LncRNAs have a role in assembly, maintenance, plasticity and abnormality of neural circuitry [100] and can act as biomarkers for diagnostic and therapeutic interventions in autism.

Human LncRNA microarray analysis on postmortem brain samples displayed over 200 differentially expressed LncRNAs in ASD [101]. The salient findings here were that the number of lncRNAs differentially expressed within control brains was greater than lncRNAs differentially expressed within autism brains (1375 lncRNAs versus 236 lncRNAs, respectively) [101]. Interestingly, almost 50% of differentially expressed LncRNAs map to within 50 Kilobases (Kb) of an annotated gene and they made a salient observation that both LncRNA and mRNA transcriptome appear to be differentially expressed within control brains compared to ASD brains [101]. This finding lends support to the functional magnetic resonance imaging (fMRI) studies, which indicate failure in development of cortical networks in high functioning individuals with autism [102] and the large difference observed in regional cortical differential gene expression between ASD cases and controls [103].

Another LncRNA MSNP1AS was highly overexpressed in the postmortem cerebral cortex of individuals with ASD [53]. This study also shows that *MSNP1AS* is antisense to and can bind moesin (*MSN*) transcript, and overexpression of *MSNP1AS* causes a decrease in moesin protein that regulates neuronal architecture [53]. DeWitt and colleagues [104] showed MSNP1AS knockdown in human neuronal progenitor cells disrupted the expression of 318 genes, many of which are involved in chromatin organization and immune response. It has been shown that LncRNA *LOC389023*, positioned in chromosome 2q14.1, within the *DPP10* gene [105], can regulate specific voltage-gated potassium channels and alters their expression which controls neuronal functions. Few studies dealt with LncRNAs in ASD using blood [106,107]. Peripheral leukocytes from ASD subjects identified thirteen synaptic LncRNAs (9 up-regulated and 4 down-regulated) and 19 synaptic mRNAs (12 up-regulated and 7 down-regulated) [106], which are important in synaptic vesicle transportation and cycling in ASD. Some of the LncRNA include STX8, SYP-AS1, STXBP5-AS1, BDNF-AS, SHANK2-AS3, and HOXA- [106]. In another study, out of three LncRNAs (*NEAT1, PANDA**, and TUG1*), only *NEAT1* and *TUG1* showed significant upregulation in ASD cases [107].

Epigenetic mechanisms can regulate cellular processes including differentiation, apoptosis, and metabolism; and clinical implications of dysregulated ncRNAs (miRNA and LncRNA) in ASD could have potential for personalized therapies as diagnostic biomarkers or indicators of prognosis.

## 4. Knockout Models

The generation of a relevant disease specific knockout model for autistic traits/behaviors has been difficult because of the genetically heterogeneous nature of ASD. A number of rodent models (knockout, humanized knock-in mice, and Cre-loxP) for rare variations (de novo and CNV) detected in ASD have been used to understand the etiology of ASD [108,109,110,111]. Cadps2-knockout model showed decreased brain-derived neurotrophic factor (BDNF) release from neocortical and cerebellar neurons [112]. *CADPS2* has an important role in BDNF secretion, as it regulates the exocytosis of synaptic and dense-core vesicles in neurons. [112]. A conditional knockout model of TAOK2 gene using Cre-loxP system [113], showed gene dosage- dependent abnormalities in brain size, neural connectivity, and reduced excitatory neurotransmission. Three de novo mutations in *TAOK2* gene has been reported by WGS and WES approach, and functional analysis show these mutations differentially impact kinase activity, dendrite growth, and spine/synapse development [113].

Human in-vitro models using induced pluripotent stem cell (iPSC)-derived neurons and astrocytes are particularly valuable for ASD studies and there are a number of protocols in use to differentiate iPSC into neurons [113,114]. Jinek and colleagues [114] developed the CRISPR/Cas9 technology based on the RNA-programmed deoxyribonucleic acid (DNA) cleaving activity of the Cas9 enzyme and this revolutionized the genome engineering community to enable efficient site-specific genome editing to generate isogenic cell lines from iPSCs [110,115,116]. The application of this technology to ASD-related genes can help us understand signaling pathways in ASD; but to date limited isogenic cell lines from iPSCs have been generated [117,118,119,120].

Mutations in the *CHD8* gene are associated with ASD [121], and WES studies in ASD lend support to this finding [64]. CRISPR/Cas9 technology was used to knockout one copy of *CHD8* in a control iPSC line [118] and *CHD8*+/− iPSC model showed that *CHD8* regulates multiple genes implicated in ASD pathogenesis. Identified genes influence brain volume and are involved in cell communication, extracellular matrix and neurogenesis that are critical for brain development [118]. Furthermore, RNA-seq was carried out on *CHD8*+/− and isogenic control (*CHD8*+/+) cerebral organoids [119] and showed upregulation of the *DLX* gene family, which encodes a transcription factor involved in GABAergic interneuron differentiation. Additionally, genes expressed in *CHD8* mutant and wild-type organoids were enriched for pathways involved in neurogenesis, neuronal differentiation, forebrain development, Wnt/β-catenin signaling and axon guidance [119]. The developmental disorder locus 1q21.1 is present in autism [122]. Notch signaling is central to brain development and, NOTCH2NL deletion accelerates differentiation into cortical neurons [120]. NOTCH2NL is an important neurodevelopmental gene, where duplications are associated with macrocephaly and autism [120]. CRISPR/Cas9 gene editing strategy was used to investigate the effect of 10 additional ASD-related genes (i.e., *AFF2/FMR2*, *ANOS1*, *ASTN2*, *ATRX*, *CACNA1C*, *CHD8*, *DLGAP2*, *KCNQ2*, *SCN2A* and *TENM1*) on neuronal function [117]. Associations between genetic variants and phenotypes were observed for KO of either of the genes *AFF2*/*FMR2*, *ASTN2*, *ATRX*, *KCNQ2* and *SCN2A* with significantly reduced spontaneous excitatory postsynaptic current frequencies in iPSC-derived excitatory neurons. The results from KO studies show that ten ASD-risk genes of varying function have similar transcriptional rewiring and electrophysiological phenotypes in human iPSC-derived glutamatergic neurons [117].

The above outlined studies with CRISPR/Cas9 system, show this RNA-based genome-editing tool has the potential to engineer pre-clinical models to understand the molecular and cellular pathways in ASD, as it allows site-specific genome editing to generate isogenic cell lines from iPSCs. Future, gene editing strategies will help us understand better, the events at the molecular level of synaptic dysfunction, calcium signaling, chromatin remodeling and transcriptional regulation.

## 5. Endophenotypes

Gottesman and Gould [123] define endophenotypes as “measurable components unseen by the unaided eye along the pathway between disease and distal genotype, have emerged as an important concept in the study of complex neuropsychiatric diseases. An endophenotype may be neurophysiological, biochemical, endocrinological, neuroanatomical, cognitive, or neuropsychological (including configured self-report data) in nature”. ASD associated with comorbidities such as aggression, intellectual disability, anxiety, epilepsy, and sleep disorders. As ASD is clinically heterogeneous, endophenotypes of ASD can serve as measurable markers as they will represent a more homogeneous spectrum of the disease.

The most reliable endophenotypes in ASD fall in following clinical subgroups: hormonal, biochemical, immunological, morphological, neurophysiological/neuroanatomical, neuropsychological, and behavioral [124]. Studies which have described endophenotypes of ASD are for hyperserotonemia [125], abnormalities of electroencepalography (EEG) [126], neuroimaging [127], head circumference [128], immunological [129], and language delay [130]. Moreover structural brain region pathophysiologies also constitute endophenotypes, for example, white matter [131], and grey matter [132]. Endophenotypes can help in overall stratification of patients in various sub-groups of clinical entities for studies focused on quantitative genetic, targeted profiling, WES or WGS. Both, genetics and endophenotypes data are very complex and designing such future studies on endophenotype stratified samples will yield molecular subgroups that are potentially linked to different ASD clinical characteristics, and will aid in better patient care.

## 6. Pharmacogenomics

Advances in the field of pharmacogenomics (PGx) will help realize the objective of personalized medicine in ASD. The majority of PGx studies to date have focused on commonly utilized medication classes for ASD such as antipsychotics, antidepressants and stimulants [24,133]. ASD is a lifelong condition, and there is no pharmaceutical intervention which can fully alleviate ASD symptoms. A disruption in cortical development [55,111] is common to most patients with ASD). The core feature of disrupted social communication is unlikely to protect against psychiatric illness. While the etiologies of ASD are manifold, it is remarkable that in selected cases psychotropic medicationscan improve the quality of life. The goal of PGx is to give the right psychotropic at the right dose and the dosing guidelines for the neurotypical population only gives approximations when using psychotropic medications in autistic patients. There are only two medications with FDA approval for treating autistic patient’s irritability, risperidone and aripiprazole. metabolized by *CYP2D6* and *CYP3A4*. SNPs in *CYP2D6*, and *CYP3A4* genes that encode enzymes responsible for risperidone and aripiprazole metabolism contribute substantially to interindividual variability in dose requirement. The genetic polymorphisms of *CYP2D6* are contributing to predicting dosing for risperidone and aripiprazole [133,134,135] and are becoming important to improve the clinical outcome in autism. Both drugs are atypical antipsychotic agents and with significant long-term considerations such as weight gain and movement disorders [110]. Pre-emptive PGx testing for *CYP2D6* and *CYP3A4* genotypes in ASD vulnerable populations can inform clinicians to dosing and treatment options for risperidone and aripiprazole. The other agents commonly used include selective serotonin inhibitors, mood stabilizing antiepileptic agents such as lamotrigine, stimulants as well as alpha-receptor antagonist, and anxiolytic agents.

Non-core symptoms found in patients include irritability, aggressive behaviors, repetitive behaviors and mood related conditions. The complexity in identifying comorbid psychiatric disorders is often limited by the limitations in communications. This conditions like most of the rest of psychiatry are merely behavioral diagnoses and do not predict a molecular cause. The use of pharmacogenomics testing in the general population is established. While it also does not suggest a molecular cause PGx testing provides supportive data to assist with treatment decisions. While there are a few PGx studies with ASD patients in general they follow the more established findings seen in neurotypical psychiatric patients. Additionally, most of the studies that have been done suffer from low numbers of subjects and limited genetic data. Some focus primarily on pharmacokinetic data while others have limited pharmacodynamics data relating to receptors and transporters. Our adult psychiatry clinic (UAMS) uses commercial laboratories with limited data to perform PGx assays. It results in helping identify treatment refractory cases as well as in patients with medication intolerability.

## 7. Conclusions and Future Perspective

ASD is highly heritable and heterogenous group of neurodevelopmental disorder characterized by alterations in social interaction, communication, and repetitive behaviors. The above listed studies show genetic heterogeneity in ASD. Large cohort studies using WGS, miRNA, LncRNA has the potential for biomarker development. Carefully designed studies incorporating PGx and endophenotype stratified patient population will help further define the unique genetic underpinnings involved in cellular and molecular pathways of ASD and help in personalized therapies. Finally, the introduction of genome editing technology, CRISPR/Cas9 will help us better understand the impact of the mutations in the neural circuitry which is important in molecular dysregulation of ASD.

## Data Availability

Not applicable.

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
