# Peer review of "Molecular Dysregulation in Autism Spectrum Disorder"

_jpm, 2021, doi:10.3390/jpm11090848_

Round 1

Reviewer 1 Report

Thank you for the opportunity to review the manuscript “Molecular dysregulation in Autism Spectrum Disorders” by Pritmohinder S. Gill et al.

Autism spectrum disorders become more frequently diagnosed and the review gives relevant information about current perspectives in molecular background, as well as in diagnostic procedures in ASD. Please find my comments and suggestions for the Authors.

  1. I would recommend adding a Table with a summarize of different genes hypothetically involved in ASD diagnosis. It would be valuable for readers and make the text more easily to follow.
  2. Line 265: “Patowary and colleagues [65] looked at 26 families with affected first cousins from the NIMH repository, to identify genes with private gene disrupting and missense variants of interest (VOI).” I cannot see the explanation for “NIMH repository”.
  3. “This RNA-based genome-editing tool has the potential to engineer pre-clinical models to understand the molecular and cellular pathways in ASD, as it allows site-specific genome editing to generate isogenic cell lines from iPSCs. Future, gene editing strategies will help us understand better, the events at the molecular level of synaptic dysfunction, calcium signaling, chromatin remodeling and transcriptional regulation.”- this part of the text is unclear, please revise the meaning of the paragraph and add the explanation about “RNA-based genome-editing tool”.
  4. Line 506: “Studies which have described endophenotypes of ASD are for hyperserotonemia [124], abnormalities of electroencepalography (EEG) [125], neuroimaging [126], head circumference [127], immunological [128], language delay [129], white matter [130], ASAD-Polygenic score [131] and grey matter [132].”- what does “white matter and grey matter” mean in this context?
  5. Line 506: “PGx studies [134-137]”- I cannot see the abbreviation’s explanation.
  6. Line 558: “authoritative reviews [16, 29,31, 108-109, 138-144].”- I would recommend either write what relevant conclusions come from these papers or delete this sentence and the ref.
  7. Conclusions- I would strongly recommend adding the relevant more specific information about the new findings described in the manuscript.
  8. There is no Author Contributions, Acknowledgements etc. sections
  9. Overall, in my opinion the manuscript should be more concise. Please reconsider the number of references, as the list is definitely too long.

Author Response

Thank you for your constructive suggestions and comments. We all very much valued this input to improve the manuscript entitled "Molecular dysregulation in Autism Spectrum Disorders". Following are our responses to your suggestions and comments. These are shown line by line in the word doc and the manuscipt.

Thank you.

  1. I would recommend adding a Table with a summarize of different genes hypothetically involved in ASD diagnosis. It would be valuable for readers and make the text more easily to follow.

(Supplementary Table showing associated genes has been added. Supplementary Table S1; and using DAVID informatics program we also show some genes involved in cellular components; molecular function and biological processes Table S2). See lines110-112.

  1. Line 265: “Patowary and colleagues [65] looked at 26 families with affected first cousins from the NIMH repository, to identify genes with private gene disrupting and missense variants of interest (VOI).” I cannot see the explanation for “NIMH repository”.

For clarity of the sample source, I have added the link to NIMH repository. See lines 336-339

  1. “This RNA-based genome-editing tool has the potential to engineer pre-clinical models to understand the molecular and cellular pathways in ASD, as it allows site-specific genome editing to generate isogenic cell lines from iPSCs. Future, gene editing strategies will help us understand better, the events at the molecular level of synaptic dysfunction, calcium signaling, chromatin remodeling and transcriptional regulation.”- this part of the text is unclear, please revise the meaning of the paragraph and add the explanation about “RNA-based genome-editing tool”.

This RNA-based genome editing tool is the one using CRISPR/Cas9. I have added that and made it more clear in the section and conclusion. See lines 543,574

  1. Line 506: “Studies which have described endophenotypes of ASD are for hyperserotonemia [124], abnormalities of electroencepalography (EEG) [125], neuroimaging [126], head circumference [127], immunological [128], language delay [129], white matter [130], ASAD-Polygenic score [131] and grey matter [132].”- what does “white matter and grey matter” mean in this context? .

Moreover, structural brain region patho-physiologies also constitute endophenotypes, for example, brain white matter and grey matter referenced here. See lines 596-597

  1. Line 506: “PGx studies [134-137]”- I cannot see the abbreviation’s explanation.

In the beginning of this section, it shows now Pharmacogenomics (PGx). See line 609

  1. Line 558: “authoritative reviews [16, 29,31, 108-109, 138-144].”- I would recommend either write what relevant conclusions come from these papers or delete this sentence and the ref.

Thank you for your suggestion, I have deleted these from the references. See lines 646

  1. Conclusions- I would strongly recommend adding the relevant more specific information about the new findings described in the manuscript.

Thank you for your suggestion, and I have added this information on WES/WGS, miRNA and LnCRNA and genome editing methods in the conclusions. See lines 655-666

  1. There is no Author Contributions, Acknowledgements etc. sections

I have added this information in the manuscript

  1. Overall, in my opinion the manuscript should be more concise. Please reconsider the number of references, as the list is definitely too long.

Thank you for your suggestion-I have deleted some references. See lines 854, 1030, 1042-1063

Reviewer 2 Report

Dear authors and editors, attached please find a file with specific comments, but below please find a summary of my impression. I will note that as I am not an expert in ASD genetics, rather an autism researcher with a strong background in genetics, my comments do not relate to the accuracy of the review or the findings that it may or may not present. They relate to the clarity and flow of the content, and to the main take-home message it conveys.

My key impression is this: The paper attempts to cover a very large topic of genetic, epigenetic, and knnockout studies - with the explicit aim of "...summarize the findings ...to understand the underlying molecular mechanisms of ASD". The paper indeed covers a lot of literature on the above topics, but it does so in a way that does not leave the reader with a better understanding of the molecular mechanism of ASD, nor does the paper dedicate a concluding paragraph to bring all the massive information together. This requires major work, though interestingly, the clarity and presentation of information become extremely improved starting in the middle of pg 6 (section 2.4) - so work is required to make the first half of the article similarly clear.

My main comments are (see elaboration in the attached file):

1) The specific contribution of this review over many other similar reviews in the field is unclear and should be emphasized, both explicitly but also in the take-home messages that the paper wishes to deliver - help the reader better understand the shared mechanisms behind all the various findings. I recommend adding such a paragraph before the concluding section.

2) Critical comment (relevant until section 2.4, line 225 - then the ms dramatically improves): The text, while describing many different studies, is in many places unclear and lacks a narrative. This manifests in two ways:

a) global across the MS: The reader most often feels like a list of findings is thrown at him/her, without opening or concluding sentences. I suggest the authors ask themselves what is the main point of each paragraph - and whether the text indeed supports that point. This is said in a kind way - it is clear that a lot of evidence has been collected, but what is missing is to present them in a way that is coherent and clear.

b) within paragraphs: often enough the sentences are not well structured, or have no "point". It is also common in the paper that sentences appear with no connection or link between them and the previous sentence, or even between two half of the same sentence! This contributes to the lack of coherence mentioned above.

c) I am not sure who is the target audience, but unless it is specific for geneticists - many terms and specific terminology must be clarified and defined in order for the reader to be able to interpret the findings and build some understanding of them, as the text moves forward.

**As mentioned above, there is a dramatic change in writing quality from the section: "2.4. Whole Exome and Genome sequencing". Starting there, the paragraphs are doing a great job in summarizing the evidence in a coherent way which helps the reader link between different studies and clearly understand the take-home message of each paragraph.

Best of luck.

Author Response

Thank you for your constructive suggestions and comments. We all very much valued this input to improve the review manuscript titled Molecular dysregulation in Autism Spectrum Disorders. Following are our responses to your suggestions and comments.

Thank you,

  1. I have tried to delete some of the information in sections- 1, 1.1, 1.2 and 2, 2.1, 2.2, 2.3 and also per your suggestion attempted to concise the information at the end of each section.
  2. Your embedded comments has been responded in manuscript per lines shown: line 44, on heritability; Reference missing lines 45;line 94 deleted; Line 222; After consulting with my three co-authors, who are clinicians and see autism patient’s, they suggest we need to keep the sub-headings in section 2 as they are for the sake of studies which define those sub-headings.

  1. Below show our responses to your specific comments and suggestions.

  1. I am missing (here or later in the text) the contribution of this review over many other reviews which I am sure exist on the same topic. I recommend highlighting your novelty and contribution.

(I have changed the abstract and conclusions to highlight the novel sections discussed in this review, for example, whole exome sequencing (WES), Whole genome sequencing (WGS), non-coding RNA (miRNA and LncRNA) and genome editing examples). Lines 22-27.

  1. unclear sentence

(I have deleted it) please see lines 44-48.

  1. unclear: genetic relatedness between the kids? or if they had a relative with autism?

I have deleted it and repharased it to make it clear that risk of autism 10 fold, if a full sibling has the diagnosis ) lines 87-90.

  1. What is characterized? This sentence should be clearer

I have repharased it-It is ASD line 114

  1. For supporting such a sentence, the authors should provide a definition of what are rare versus common variants earlier in the text

(this is targeted to clinicians and geneticists, I think making it too simple will lose its focus). However, I have added its explanation-lines 79-80.

  1. Elaboration needed

(see line 120-which shows weaknesses as probe design, high cost and accuracy issues)

  1. An interpretation of this is warranted: what does this mean to a non-geneticist/scientist

This is targeted to clinicians and geneticists-who have good understanding of laborious classical linkage studies), I have tried to simplify. See lines 138-140

  1. better to rephrase or elaborate to tie to the previous sentence

This now reads: Genotyping of significant association results in additional families revealed a SNP lines 140-143.

  1. This opening sentence is great - because it feels like a beginning of an elaboration on these genetic abnormalities. However, it isn't. That causes a lot of confusion later on in the paragraph.

Thank you for your comment, I have tried to rearrange and explain better in the text) See lines 165-176.

The link between the first half of this sentence (plus previous sentence) and between the next half of it is unclear.

 Thank you for your comment, I have tried to rearrange and explain better in the manuscript. See lines as176-212

  1. Define

(Please see lines 79-81)

  1. SNV

 (Yes it is single nucleotide variation (SNV).

  1. what does that mean for the reader? Interpretation and relation to the main idea is missing

Now reads: CNV loci exhibited the highest neuro-functional networks. See lines 201-202.

(authors - created a CNV map for ASD using 9337 cases and 5650 controls and statistically marked genomic

regions and  performed gene function enrichment analysis for CNV genes and built functional networks, pathways and examined their expression in brain tissues).

  1. Format

OK

  1. Probabaly in conclusion

Thank you for suggestion

  1. Good start
  2. great interpretation of the evidence presented, also link between sentences here and narrative of the paragraph is much better
  3. good summary of findings
  4. These past few paragraphs are doing a great job in summarizing the evidence in a coherent way which helps the reader link between different studies and clearly understand the take-home message of each paragraph
  5. We thank the reviewer for their encouragement).
  6. BDNF

(Thank you, I have corrected per your suggestion). See lines 528-529

  1. shouldn't this be described as a part of the previous paragraph? as an extension of it

Thank you for your suggestion.

Actually where your comment is in paragraph, I am giving examples of ASD genes using this technology. I have combined with the previous paragraph as suggested.

Round 2

Reviewer 2 Report

The authors have provided sufficient revisions to address the concerns I brought up and I recommend their paper to be accepted after English typos and grammar mistakes will be checked more thoroughly and corrected. Great job in making the first half of the paper more readable, comprehensive and conclusive.